# The Development of a Method for Obtaining *Tripleurospermum inodorum* (L.) Sch. Bip. Herb Extract Enriched with Flavonoids and an Evaluation of Its Biological Activity

**DOI:** 10.3390/plants13121629

**Published:** 2024-06-13

**Authors:** Anna Marakhova, Vera Yu. Zhilkina, Alexander Elapov, Nadezhda Sachivkina, Alexander Samorodov, Kira Pupykina, Irina Krylova, Parfait Kezimana, Anastasia M. Stoynova, Raja Venkatesan, Alexandre A. Vetcher

**Affiliations:** 1Institute of Biochemical Technology and Nanotechnology, Peoples’ Friendship University of Russia n.a. P. Lumumba (RUDN University), 117198 Moscow, Russia; marakhova_ai@pfur.ru (A.M.); zhilkina_vyu@pfur.ru (V.Y.Z.); alexandrea@ya.ru (A.E.); kezimana-p@rudn.ru (P.K.); stoynova-am@rudn.ru (A.M.S.); 2Department of Microbiology V.S. Kiktenko, Medical Institute, Peoples’ Friendship University of Russia n.a. P. Lumumba (RUDN University), 117198 Moscow, Russia; sachivkina@yandex.ru; 3Department of Pharmacy, Bashkir State Medical University, 450008 Ufa, Russia; avsamorodov@bashgmu.ru (A.S.); pupykinaka@gmail.com (K.P.); i.krylova16@yandex.ru (I.K.); 4School of Chemical Engineering, Yeungnam University, Gyeongsan 38541, Republic of Korea

**Keywords:** *Tripleurospermum inodorum*, flavonoids, *S. aureus*, *E. coli*, *C. albicans*, anti-inflammatory activity, anticoagulation activity

## Abstract

The development of new drugs derived from plant sources is of significant interest in modern pharmacy. One of the promising plant sources for introduction into pharmaceuticals is *Tripleurospermum inodorum* (L.) Sch. Bip., also known as *Tripleurospermum perforatum* (Merat.) M. This plant has been shown to possess various biological activities, including anti-inflammatory, antimicrobial, and antimycotic activities, among others. However, a review of the current literature reveals a paucity of studies investigating the chemical composition of the herb *Tripleurospermum inodorum* (L.) Sch. Bip. This study presents the development of a method for obtaining an extract of the herb *Tripleurospermum inodorum* (L.) Sch. Bip. enriched with flavonoids, harvested before flowering and butonization. This study focused on determining the optimal conditions for extraction, including the concentration of the extractant (ethanol), extraction time, raw material/extractant ratio, extraction frequency, complexation reaction time, amount of aluminum chloride solution, and amount of diluted acetic acid. The results indicate that herbs harvested during this specific period exhibited a higher flavonoid content compared to those collected during butonization and flowering. Moreover, this study demonstrated that the flavonoid content could exceed 7% mg REq/100 g D.W. through a one-hour extraction process. Furthermore, the flavonoid content was found to be 7.65 ± 0.03 mg REq/100 g D.W. following a three-minute ultrasound-assisted extraction process, followed by thermal extraction. A qualitative analysis identified a variety of phenolic compounds in the extract, such as chlorogenic acid, 5-*O*-p-coumaroylquinic acid, 1-*O*-p-coumaroylquinic acid, luteolin-7-glucoside, quercetin-3-glucoside, luteolin-7-rutinoside, 3,5-*O*-dicaffeoylquinic acid, quercetin-3-*O*-malonylglucoside, apigenin-7-glucoside, luteolin-3-malonylglucoside, cynarin, rhamnetin-3-(*O*-dimethyl rhamnosyl glucosylglucoside), and luteolin. Moreover, this study demonstrated the antimicrobial, anti-inflammatory, anticoagulant, anti-aggregation, and antioxidant activities of the aqueous alcoholic extract from *T. inodorum* herb (ETIH) against pathogens such as *Staphylococcus aureus*, *Escherichia coli*, and *Candida albicans*. Additionally, the extract exhibited comparable anti-inflammatory effects on diclofenac sodium. These findings contribute to the understanding of the potential pharmacological applications of the developed herb extract.

## 1. Introduction

The development of new drugs derived from plant sources presents a significant challenge in modern pharmacy and medicine due to the distinct advantages of plant remedies, including low toxicity, broad therapeutic effects, and cost-effectiveness.

*Tripleurospermum inodorum* (L.) Sch. Bip. has the potential to be developed into a pharmaceutical agent due to its biological activities, including anti-inflammatory, antimicrobial, and antimycotic properties.

*Tripleurospermum inodorum* (L.) Sch. Bip. is a plant from the family Asteraceae that is widely distributed in various countries around the world [1,2,3,4,5,6,7,8,9,10,11]. In the official medical practices of Russia and Poland, *T. inodorum* is considered an admixture in the flowers of chamomile (*Chamomilla recutita* (L.)) [12]. In certain countries, i.e., Great Britain and Canada, *T. inodorum* is classified as a weed [13,14]. The literature contains reports of the use of plants belonging to the genus *Tripleurospermum* as antispasmodic [4], emollient [14,15], anti-inflammatory [16], antiherpetic, antioxidant, anticholinesterase, and analgesic agents [17,18,19,20,21,22,23,24,25].

Although some studies have investigated the chemical composition and pharmacological properties of *Tripleurospermum inodorum* flowers, which have also been used as folk medicine, there is a lack of information regarding the herb’s chemical composition, phenolic compounds, and pharmacological action when harvested before flowering. Data on the chemical compositions of flowers of plants of the genus *Tripleurospermum*, including the contents of free organic acids, flavonoids, and tannins in the flowers of *T. inodorum*, have been reported in several studies [8,18,19,26,27,28,29]. The compositions of essential oils [30] and fatty acids [31] have been studied in detail. The qualitative composition of phenolic compounds was studied in *T. inodorum* flowers collected in Serbia, and a qualitative analysis of LC-MS/MS identified 19 different flavonoids and their derivatives, including derivatives of myricetin, luteolin, quercetin, and luteolin, among others [7,32].

Initial investigations have indicated that the *T. inodorum* herb, collected prior to flowering, contains a notable quantity of flavonoids. The plant’s phytomass at this stage of development is well developed and suitable for harvesting. A notable quantity of phenolic compounds can elicit a range of pharmacological effects, including antimicrobial, anti-inflammatory, and anticoagulant effects, among others [16,33,34,35,36,37]. Furthermore, existing data on the extraction efficiency of flavonoids from the herb before flowering are limited despite the potential demonstrated in other plant objects [38,39,40,41,42,43,44,45,46,47,48,49]. 

Therefore, this study aims to address the aforementioned gaps by performing a quantitative determination of the phenolic compounds and establishing the chemical composition and biological activity of the herb when harvested before flowering. Moreover, this study aims to investigate the potential for enhancing the efficiency of flavonoid extraction from the *T. inodorum* herb, offering insights that could be valuable for its pharmaceutical utilization.

## 2. Results

The results of the selection of conditions for the extraction of flavonoids from the herb and conditions for the reaction of complexation of flavonoids with aluminum chloride are presented in Table 1.

The obtained data indicate that the optimal extraction conditions are as follows: the use of ethanol at a concentration of 70%, a ratio of herbal raw materials and extractant in a mass/volume ratio of 1:150, and an extraction time of 40 min.

The kinetics of the complex formation of flavonoids from the herb *T. inodorum* with aluminum chloride were subsequently investigated with the objective of determining the optimal time for the reaction. The results demonstrate that the maximum optical density is reached 30 min after the start of the reaction (Figure 1).

The results indicate that a solution of 2 mL of aluminum chloride was sufficient to complete the reaction (Figure 2).

Subsequently, the potential of fractional extraction to enhance the yield of flavonoids was investigated. To this end, a sample of the raw material, weighing approximately 1 g (with an exact weight), was poured into 100 mL of 70% ethanol and extracted by heating for 30 min, as indicated in the Section 4. Thereafter, the extract was decanted and filtered into a 250 mL volumetric flask. The initial quantity of the raw material was replenished with a fresh portion of 70% alcohol, with a volume of 100 mL, and extraction was conducted under the same conditions for a further 30 min. After cooling, the extract was filtered and combined with the first portion of the extract, and the volume was adjusted to the mark of 70% alcohol. It was demonstrated that the double extraction method resulted in a significant increase in the yield of flavonoids from the *T. inodorum* herb, with an average yield of 5.68% increasing to 7.21% REq/100 g D.W. The data on the content of total flavonoids in the *T. inodorum* herb, as a function of the growing season, are presented in Figure 3A. The data on the content of total flavonoids in the *T. inodorum* herb, as a function of the year, are presented in Figure 3B.

Figure 3A illustrates that the highest concentrations of antioxidant and anti-inflammatory compounds, specifically flavonoids, are found in the grass harvested before flowering and butanization. Consequently, subsequent studies were conducted on the *T. inodorum* herb harvested before flowering and butanization. The subsequent objective was to ascertain the stability of the flavonoid content in the grass over time. The data on the content of flavonoids in grass harvested in May 2020, 2021, 2022, and 2023 are presented in Figure 3B. The typical absorption spectrum of the flavonoid complex with aluminum chloride is presented in Figure 4 (2023 sample).

The data indicate that the total flavonoid content is at least 7% regardless of the year of harvesting.

Consequently, when incorporating this botanical material into the pharmacopoeia, it is recommended that standardization be conducted in accordance with the indicator “the amount of flavonoids in terms of rutin”, with a minimum value of 7% REq/100 g D.W.

The data on the content of total flavonoids in the *T. inodorum* herb harvested before flowering and butanization during extraction under the influence of ultrasound are presented in Figure 5.

The results indicate that ultrasound did not result in an enhanced flavonoid extraction efficiency compared to double extraction with heating. This may be attributed to the shorter extraction time. Nevertheless, the use of ultrasound during sonication for a duration of 90 s, under stipulated conditions, enables the extraction of flavonoids in an amount of 4.43 ± 0.01% REq/100 g D.W. In view of the established extraction parameters and the influence of ultrasound, an investigation was conducted to assess the potential of complex extraction, which combines classical extraction with heating, and the influence of ultrasound.

The use of complex extraction techniques enabled the yield of flavonoids to be increased to 7.65 ± 0.03% REq/100 g D.W. while simultaneously reducing the extraction time from 1 h to 33 min. A reduction in the extraction time allows for the optimization of an analysis and the implementation of cost-effective procedures.

Based on the mass spectra obtained during ionization in positive and negative modes, molecular ions (M + H^+^, M + Na^+^, and M − H^−^) were proposed, the molecular weight of the component was estimated, and the structure of the aglycone included in the composition was suggested based on fragmentation and a UV spectra analysis of the molecule. Due to the large number of components, data were collected for peaks with an area of at least 1% of the sum of all areas at 260 nm. The data obtained are presented in Table 2 and Figure 6.

The results are presented in Table 3, while the data on the anticoagulation, anti-aggregation, antioxidant, and anti-inflammatory activities are presented in Table 3, Table 4, Table 5, Table 6 and Table 7, respectively.

For studies on biological activity, a lyophilized form of the ETIH was used, which was dissolved in distilled water in a ratio of 1:10 (ETIH solution).

## 3. Discussion

A method was developed for the quantitative determination of the amount of flavonoids in the *T. inodorum* herb. The development process involved the selection of several key variables, including the extraction time, the ratio of raw materials to extractant, the concentration of ethyl alcohol, the optimal amount of complexing agent (aluminum chloride), and the complexing agent time. It has been shown that the use of a double extraction process results in a twofold increase in the yield of flavonoids from the *T. inodorum* herb. The optimal extraction technique for ETIHs was identified through the use of ultrasound at a frequency of 22 kHz and a power of 90 W for 3 min, followed by heating under reflux for 30 min. The extractant used was 70% ethanol. This extraction method has been demonstrated to result in a significant increase in the yield of flavonoids compared to the data in the literature [26,27,32].

It was found that the herb harvested prior to flowering contains the highest concentration of flavonoids and is the most promising subject for further study.

A notable absence of studies in the literature exists regarding the chemical composition of the herb during and after flowering. The majority of existing research has focused on flowers or herbs during the flowering stage. This is evidenced by studies [7,11,14,15,16,17,18,19,25,27,28,31,32]. A qualitative analysis of phenolic compounds revealed the presence of 13 distinct phenolic compounds in ETIHs, with the peak area representing at least 1% of the total area at 260 nm. This includes compounds such as chlorogenic acid, 5-*O*-*p*-coumaroylquinic acid, 1-*O*-*p*-coumaroylquinic acid, luteolin-7-glucoside, quercetin-3-glucoside, luteolin-7-rutinoside, 3,5-*O*-dicaffeoylquinic acid, quercetin-3-*O*-malonylglucoside, apigenin-7-glucoside, luteolin-3-malonylglucoside, cynarin, rhamnetin-3-(*O*-dimethyl rhamnosylglucosyl glucoside), and luteolin. A spectrophotometric method was employed for the quantitative determination of total flavonoids, and the results indicate that the content of total flavonoids, when expressed in terms of rutin and absolutely dry raw materials, exceeded a minimum of 7% across samples collected in different years. This is a high flavonoid content compared to that observed in flowers of *T. inodorum*, as presented in several studies [6,7,27,32].

Regarding biological activity, the results of our study indicate that ETIH exhibits significant activity against *Staphylococcus aureus* and inhibits the growth of *Escherichia coli* and *Candida albicans* compared to the control, a 70% ethanol solution. Given the high content of flavonoids in ETIH revealed in our study and the findings from other studies [2,16,36,37], we hypothesize that this group of compounds is responsible for the antibacterial effect.

In terms of its impact on the hemostasis system, our data indicate that ETIH exerts a minimal effect on the plasma component, as evidenced by a change in the indicator of the internal blood coagulation pathway, specifically an increase in the activated partial thromboplastin time (APTT). Heparin sodium prolongs the APTT by 20.3%, while the median prolongation of the APTT for *T. inodorum* is 7.1%. Nevertheless, at the studied concentration, the ETIH did not affect the concentration of fibrinogen or the prothrombin time (PT). Given these results, it is plausible to suggest a potential tendency towards anticoagulation activity manifesting with ETIH.

It was established that the ETIH exhibits weak anti-aggregation activity. The data obtained from the ETIH indicate that it, like the comparison drugs, reduces the maximum amplitude of platelet aggregation. However, this reduction is 6.5% for ETIH, 13.7% for acetylsalicylic acid, and 48.4% for pentoxifylline, which indicates the influence of the drugs on platelet aggregation. ETIH increases the latent period of platelet aggregation by 4.2%. In comparison, pentoxifylline increases this period by 32.4%, while acetylsalicylic acid reduces this indicator by 2.1%. The studied raw material of *T. inodorum* exhibited a significant reduction in the rate of platelet aggregation, with a 9.6% decrease, which is comparable to the reference drug, acetylsalicylic acid, with a 10.5% reduction, but differed from that of the reference drug, pentoxifylline, which had a 34.9% reduction. In terms of the time taken to reach maximum amplitude, the *T. inodorum* indicators were comparable to acetylsalicylic acid (9.6% and 10.5%) but differed from the values of pentoxifylline (32.1%). In the presence of the studied ETIH, as well as the reference drug (acetylsalicylic acid), platelet disaggregation was not observed in contrast to the other reference drug (pentoxifylline).

The results also show that the ETIH exhibits antioxidant activity in the model system for the generation of reactive oxygen species (I) and lipid peroxidation (II). However, in comparison with ascorbic acid, the level of antioxidant properties exhibited by the ETIH is inferior. In the LPO system, the AOA of ETIH and the reference drug were observed, with ascorbic acid reducing the rate of LPO by 78.1% and ETIH by 32.6%. In the second model system, ascorbic acid reduced the generation of AOS by 84.5%, and *T. inodorum* by 16.1%.

Thus, the experimental results establish that under in vitro conditions, an aqueous alcoholic extract exhibits weak anticoagulation, anti-aggregation, and antioxidant activities. ETIH had quite high antioxidant activity when compared with other plant extracts [50].

Regarding anti-inflammatory activity, this study revealed that ETIH exhibited a 66.67% inhibition of edema at the peak of its development (4 h) compared to the control group, which demonstrated a 72.22% inhibition of the exudative phase of inflammation. By the end of the experiment (24 h), the severity of the inflammatory reaction decreased and the ETIH showed results comparable to the comparison drug. The anti-inflammatory effect was comparable to that of the comparison drug, diclofenac. Nevertheless, plant extracts may offer several advantages over non-steroidal anti-inflammatory drugs, including the potential for long-term use without adverse effects on the gastrointestinal tract

Previous studies have examined the biological activity of plants belonging to the genus *Tripleurospermum* [4,14,15,16,17,18,19,20,21,22,23,24,25], as well as the individual groups of biologically active substances included in their composition [19]. However, the biological activity of the extract of *Tripleurospermum inodorum* herb harvested before flowering was established for the first time in this study.

## 4. Materials and Methods

### 4.1. Reagents

The following reagents were used: ethanol (Scharlab, Barcelona, Spain); acetic acid (Merck, Darmstadt, Germany); aluminum chloride (Merck, Darmstadt, Germany); rutin (NIST, Gaithersburg, MD, USA); methanol (Merck, Darmstadt, Germany); acetonitrile (Merck, Darmstadt, Germany); formic acid (Merck, Darmstadt, Germany); meat peptone agar (MPA) (HiMedia, Mumbai, India); Endo medium (HiMedia, Mumbai, India); Sabouraud agar medium (BioMerieux, Marcy l’Etoile, France); luminol (5-amino-2,3-dehydro-4-phthalazinedione) (Serva, Heidelberg, Germany); phosphate buffer (Sigma-Aldrich, Steinheim, Germany); KH_2_PO_4_ (neoFroxx, Einhausen, Germany); KCL (Reagecon, Ireland); sodium citrate (neoFroxx, Einhausen, Germany); KOH (Reagecon, Dublin, Ireland); ferrous sulfate (HiMedia, Mumbai, India); 3,7-dimethyl-1-(5-oxohexyl)xanthine—pentoxifylline (Dalkhimpharm, Khabarovsk, Russia); 2-acetoxybenzoic acid—acetylsalicylic acid (Shandong Xinhua Pharmaceutical Co., Zibo, China); heparin sodium (Sintez, Saint-Petersburg, Russia); ascorbic acid (Shandong Xinhua Pharmaceutical Co., Zibo, China); unrefined filtered sunflower oil (Sigma, Kopeysk, Russia); reagent kits used at the in vitro stage—coagulation test kits produced by “Technology-Standard” (Barnaul, Russia); and inducers of platelet aggregation produced by “Technology-Standard” (Barnaul, Russia).

### 4.2. Plant Harvesting

All studies were carried out using a *T. inodorum* herb extract (ETIH). The herb was harvested at different phases of the growing season, namely prior to budding, during the budding period, and during the flowering period. The wild herbs were collected in the Domodedovo district of the Moscow region in fields and along the edges of forests. Figure 7 shows the herb harvested prior to flowering. Herbal materials were naturally dried in a dark, ventilated environment.

### 4.3. Extraction and Technique Development for Analyzing Total Flavonoid Content

The total flavonoid content was quantified by spectrophotometry following the formation of a complex with AlCl_3_. A number of variables were considered in the experimental design, including the concentration of the extractant (ethanol), the extraction time, the ratio of raw material to extractant, the frequency of extraction, the reaction time for complexation, the quantity of aluminum chloride solution, and the amount of diluted acetic acid.

A sample of the herbal raw material, with a mass of 1 g (exactly weighed), was placed in a round-bottomed flask with a capacity of 250 mL and filled with ethanol in ratios of 1:50, 1:100, 1:150, and 1:200 with concentrations of 20, 40, 50, 60, 70, and 95%. The flask was connected to a reflux condenser and extracted on a hotplate with a closed spiral for 20, 30, 40, 50, and 60 min after boiling. The resulting extract was cooled, filtered through a cotton gauze filter into a 250 mL volumetric flask, and diluted to the mark with ethanol of the appropriate concentration (solution A). An aliquot of solution A, equal to 1 mL, was transferred to a 25 mL volumetric flask. Subsequently, 0.2 mL of acetic acid 30% was added, followed by 0.5, 1, 2, 3, 4, and 5 mL of 2% aluminum chloride solution. The flask was then adjusted to the mark with ethanol of the appropriate concentration and mixed. At 2, 5, 10, 15, 20, 25, 30, and 35 min, the absorption spectrum was recorded in the wavelength range of 350 to 450 nanometers against the background of a reference solution. The reference solution was prepared as follows: an aliquot of solution A (2 mL) was transferred to a 25 mL volumetric flask, 0.2 mL of 30% acetic acid was added, and the solution was adjusted to the mark with ethanol of the appropriate concentration.

In parallel, a complex of rutin with aluminum chloride was prepared according to the following procedure. Approximately 0.05 g of rutin (previously dried to a constant weight at a temperature of 100–105 °C) was dissolved in 70% ethanol in a 100 mL volumetric flask (solution A). A 1 mL aliquot of solution A was transferred to a 25 mL volumetric flask. This was followed by the addition of 2 mL of a 2% aluminum chloride solution, 0.2 mL of 30% acetic acid, and dilution to the mark with ethanol. The solution was then mixed (solution B). After 40 min, the absorption spectrum was recorded in the wavelength range of 350 to 450 nanometers against the background of a reference solution [51].

The total flavonoid content in terms of rutin and absolutely dry raw materials was calculated using the following formula:(1)$$X=\frac{A\times m_{0}\times V\times 25\times 100\times 100}{A_{0}\times 100\times 25\times a\times m\times \left(100-W\right)}\;$$
where *A*—the optical density of the test solution at 412 nm; *A*_0_—the optical density of the complex of the standard sample rutin with aluminum chloride; *m*_0_—the mass of a sample of a standard sample of rutin in g; *V*—the extraction volume in mL; *a*—the aliquot in mL; *m*—the mass of raw material in g; *W*—the weight loss of raw materials during drying in %.

Once the optimal conditions were identified, a double extraction process was carried out, with each extraction lasting 30 min. The potential for enhancing the extraction efficiency was investigated through the use of ultrasound.

### 4.4. Ultrasound-Assisted Extraction Method for Flavonoid Analysis

A total of 1.0 g (exactly weighed) of the crushed raw materials was placed in a cell of a sonic circuit, and 50 mL of 40% ethanol was poured in. The cell was sounded for 30, 60, and 90 s in a thermostatic mode at an oscillation frequency of 22 kHz and a specific ultrasound power of 24.4 kJ for 20, 30, and 40 min. Subsequently, the extract was filtered into a 100-milliliter volumetric flask through a paper filter and adjusted to 100 milliliters to the mark with 40% ethanol (solution A of the test solution).

Two milliliters of solution A of the test solution were placed in a 25-milliliter volumetric flask. Two milliliters of aluminum chloride solution (5% in 70% ethanol) and 0.2 milliliters of acetic acid (30% solution) were added, and the volume of the solution was adjusted to the mark with 70% ethanol and mixed (solution B of the test solution).

The optical density of solution B of the test solution was measured after 20 min on a spectrophotometer in the wavelength range of 350 to 500 nanometers in a cuvette with a layer thickness of 10 mm. A reference solution was prepared by combining 2.0 mL of solution A of the test solution, 0.2 mL of 30% acetic acid, and 70% ethanol to the mark in a 25 mL volumetric flask. A further analysis was conducted in accordance with the procedures outlined in Section 4.3.

### 4.5. Extraction and Qualitative Analysis of Flavonoids by UPLC/MS

An amount of 1 g of dry sample (exactly weighed) was transferred to a 50 mL round-bottom flask, 25 mL of 70% aqueous methanol was added, and the flask was placed in a boiling water bath with reflux for 1 h. Extraction was repeated twice more with 10 mL of extractant. The resulting extracts were combined, evaporated on a vacuum rotary evaporator, and dissolved in 10 mL of 70% aqueous methanol. An amount of 2 mL of the resulting solution was placed in a centrifuge tube and centrifuged at 15,000 rpm for 5 min, and the supernatant was transferred to a vial for chromatography.

Ultraperformance liquid chromatography/mass spectrometry (UPLC/MS) was employed on a Waters Acquity chromatograph equipped with a diode array ultraviolet (UV) detector and a tandem quadrupole mass spectrometer detector (TQD) (Waters).

Mobile phase A (PF A). A mixture of water/acetonitrile (95:5) with formic acid.

Mobile phase B (PF B). Acetonitrile with formic acid.

The test solution and standard solutions (solutions of USP standard samples of identified compounds 0.05%) were chromatographed under the following conditions:-Sample volume of 5 µL;-Column, 0.21 × 15.0 cm Acuity UPLC BEH C18 (1.7 µm);-Column temperature of 35 °C;-Flow rate of 0.25 mL/min;-Gradient chromatography mode formed by mixing mobile phases A and B according to the scheme involving 0 min: %B 5, 30 min: %B 50, 32 min: %B 100, 33 min: %B 5, and 36 min: %B 5;-UV detection: 220–500 nm.

MS conditions:-MS detection in positive ion mode;-Detector parameters: capillary voltage of +3 kV; cone voltage of 50 V; capillary temperature of 450 °C; source temperature of 120 °C; drying gas flow rate of 800 L/h; gas flow rate of 50 L/h in the cone; and scanning in the mass range of 100 to 1500 units;-MS detection in negative ion mode;-Detector parameters: capillary voltage of −3 kV; cone voltage of −30 V; capillary temperature of 350 °C; source temperature of 120 °C; drying gas flow rate of 500 L/h; gas flow rate of 50 L/h in the cone; and scanning in the mass range of 100 to 1500 units.

### 4.6. Antimicrobial Activity

Studies of the antimicrobial activity of *T. inodorum* were carried out against the following microorganisms: *S. aureus, E. coli,* and *C. albicans*.

The strains used were as follows:

*Staphylococcus aureus* (B-6646);

*Escherichia coli* (B-6645);

*Candida albicans* (Y-3108).

An amount of 50 μL of a suspension of a certain type of bacteria at a concentration of 2.5 × 10^8^ CFU/mL was seeded into a “lawn” using a spatula onto meat peptone agar (MPA) (*S. aureus*) or Endo medium (*E. coli*) at a thickness of 5 mm in a Petri dish. Yeast-like fungi of the genus *C. albicans* were inoculated in the same volume as the “lawn” on Sabouraud agar medium.

Then, sterile filter paper discs soaked in aqueous alcoholic ETIHs were diffused onto the agar inoculated with cultures of microorganisms. The disks were placed at equal distances from each other and a distance of 2 cm from the edge of the Petri dish. An aqueous alcoholic ETIHs, 25 μL, was placed on the disks using an automatic pipette with sterile removable spouts. The dishes were placed in a thermostat at 37 °C for 48 h. The results were assessed by the phenomenon of delayed growth of microorganisms around the disks using a ruler, including the diameter of the disk itself. The degree of sensitivity of microorganisms to the test samples was determined by the size of the zone of no growth of microorganisms. Microbiological studies were carried out in laminar, pre-irradiated with a UV lamp for sterilization before sowing test cultures on nutrient media [52,53,54,55,56,57,58].

### 4.7. Anti-Inflammatory Activity

Experimental work under in vivo conditions was carried out on 30 outbred male Wistar rats weighing 250–300 g, which were kept in vivarium conditions in accordance with the provisions of the “European Convention for the Protection of Vertebrate Animals Used for Experiments or Other Scientific Purposes ETS N 123”. (ETS No. 123, Strasbourg, 1986) [59].

An acute inflammatory reaction (edema) was reproduced by a subplantar (under the plantar or plantar aponeurosis) injection of 0.1 mL of 2% formalin solution. Dry ETIH was administered forcibly, dissolved in water in a ratio of 1:10 at a dosage of 10 mg/kg in an amount of 1 mL daily for a week. Swelling of the animals’ paws was determined by the difference in paw diameter (in mm), which was measured with a caliper 4 and 24 h after the induction of inflammation relative to the diameter of the paw before the induction of inflammation. The reference drug (diclofenac sodium) was administered 1 h before modeling edema at a dosage of 10 mg/kg intraperitoneally. The anti-inflammatory effect was assessed by reducing the swelling of the paws in rats against the background of the studied extract relative to the control group.

The severity of the inflammatory reaction was assessed as a percentage using the following formula:(2)$$X=\frac{(A-B)\times 100}{B}\;$$
where *X* is the increase in the diameter of the paw, *A* is the diameter of the diseased paw in mm, and *B* is the diameter of a healthy paw in mm.

The intensity of inflammation was judged by the percentage of suppression of edema at the peak of inflammation. The edema inhibition index at the peak of inflammation was calculated using the following formula:(3)$$W=\frac{(X0-X)\times 100}{X0}$$
where *W* is the edema inhibition index at the peak of inflammation, *X*0 is the increase in paw diameter in the control group, and *X* is the increase in paw diameter in the study group. Comparison drug: diclofenac sodium (Grotex LLC, St-Petersburg, Russia).

### 4.8. Anti-Aggregation, Anticoagulation, and Antioxidant/Prooxidant Activity

In vitro experiments were performed on the blood of healthy male donors aged 18–24 years. The total number of donors was 11 people. Blood collection for the study of the connection to the hemostasis system was carried out from the cubital vein using BD Vacutainer^®^ vacuum blood collection systems (Becton, Dickinson and Company, Sparks, MD, USA). Blood collection was approved by the RUDN Ethical Committee. A 3.8% sodium citrate solution in a ratio of 9:1 was used as a venous blood stabilizer. All tests were performed on platelet-rich and platelet-depleted plasma. Platelet-rich plasma samples were obtained by centrifuging citrated blood at 1000 rpm for 10 min, and platelet-free plasma were obtained at 3000 rpm for 20 min. The centrifuge OPN-3.02 (OJSC TNC “DASTAN”, Bishkek, Kyrgyzstan) was used in this work.

The study of the effect on platelet aggregation was carried out using the Born method [60] on an AT-02 aggregometer (Medtech, Moscow, Russia). Adenosine diphosphate (ADP) at a concentration of 20 μg/mL and collagen at a concentration of 5 mg/mL produced by Tekhnologiya-Standard (Russia) were used as aggregation inducers. The maximum amplitude of aggregation, the rate of aggregation, the time taken to reach the maximum amplitude, and disaggregation in the presence of the studied compounds during ADP-induced platelet aggregation were assessed. During collagen-induced platelet aggregation, the latent period during which phospholipase C is activated (which leads to the formation of second messengers, resulting in the secretion of platelet granules and the synthesis of thromboxane A2) was assessed.

Determination of anticoagulation activity was carried out using generally accepted clotting tests on an optical two-channel automated blood coagulation analyzer ASKa 2-01-”Astra” (Astra, Ufa, Russia). The parameters of activated partial thromboplastin time (APTT), prothrombin time (PT), and fibrinogen concentration according to A. Clauss were studied. Reagents produced by Tekhnologiya-Standard, Ltd. (Barnaul, Russia) were used in this work.

Antioxidant properties were assessed in simple model systems that simulate the most common free radical oxidation reactions in the body and in environments in which the formation of reactive oxygen species and lipid peroxidation reactions were initiated. The registration of luminescence was carried out using a “KHLM-003” chemiluminometer (Ufa State Aviation Technical University, Ufa, Russia). Antioxidant activity was determined by the degree of chemiluminescence inhibition and recalculated as a percentage of the control. Ascorbic acid was chosen as a comparison drug. To detect reactive oxygen species, luminol (5-amino-2,3-dehydro-4-phthalazinedione) was used, which oxidizes and forms electronically excited carbonyl chromophores with a high quantum yield, resulting in a sharp increase in the intensity of the glow associated with the formation of active forms of oxygen. Chemiluminescence was recorded for 5 min. The study of herbal remedies was carried out by adding 1 mL of solution to 20 mL of the reaction mixture.

To initiate reactive oxygen species (model I), 20 mL of phosphate buffer was used with the addition of citrate and luminol. Buffer composition: 2.72 g KH_2_PO_4_, 7.82 g KCL, and 1.5 g sodium citrate C_6_H_8_O_7_Na_3_·5.5H_2_O per 1 L of distilled water. The pH value of the resulting solution was adjusted to 7.45 units of titration with a saturated KOH solution, and 0.2 mL of luminol stock solution (10^−5^ M) was added. The formation of ROS was initiated by introducing 1 mL of a 50 mM solution of ferrous sulfate.

To evaluate the effect of compounds on lipid peroxidation (model II), lipoprotein complexes were prepared from chicken yolk. The yolk was mixed with a phosphate buffer in a ratio of 1:5 and then homogenized. Chemiluminescence was initiated by adding 1 mL of 50 mM ferrous sulfate solution, which triggered the oxidation of unsaturated fatty acids that make up lipids. The intensity of the developing glow was used to judge the processes of lipid peroxidation.

To assess biological activity, the extract was added to the plasma at a rate of 5% of the volume of the reaction mixture. The anti-aggregation activity of pentoxifylline and acetylsalicylic acid is presented for a concentration of 2 × 10^−3^ M, and the anticoagulation activity of sodium heparin is presented for a concentration of 5 × 10^−4^ g/mL.

The following were used as comparison drugs:3,7-dimethyl-1-(5-oxohexyl)xanthine—pentoxifylline;2-acetyloxybenzoic acid—acetylsalicylic acid;Heparin sodium;Ascorbic acid;Unrefined filtered sunflower oil.

The following reagent kits were used for work at the in vitro stage:Coagulation test kits produced by “Technologiya-Standard”:-Tech-APTV-El-test, -Tech-Fibrinogen-test, -Techplastin-test (R).


2.Inducers of platelet aggregation produced by “Technologiya-Standard”:-ADP;-Collagen.


### 4.9. Statistical Analysis

The research results were processed using the statistical software package Statistica 10.0 (StatSoft Inc., Tulsa, OK, USA). The actual data were subjected to a test for normality using the Shapiro–Wilk test. It was demonstrated that the type of distribution of the obtained data differed from normal, thus necessitating the use of nonparametric methods in subsequent analyses. The data are presented as the median and 25th and 75th percentiles. The Kruskal–Wallis test was employed for the analysis of variance. The critical significance level, *p*, for statistical criteria was set as 0.05.

## 5. Conclusions

Our study involved the extraction of aqueous alcoholic extract from *T. inodorum* herb using ultrasound processing followed by heating with 70% ethanol as an extractant. Through the analysis of TIHs harvested before and after flowering, we observed that the pre-flowering harvest contained the highest concentration of flavonoids, suggesting that it is the most promising object of study.

Using UPLC/MS, 13 phenolic compounds were identified in the pre-flowering aqueous alcoholic extract from *T. inodorum* herb, with peak areas constituting at least 1% of the total sum of areas at 260 nm. This comprehensive analysis provided valuable insights into the phenolic composition of the aqueous alcoholic extract from *T. inodorum* herb, indicating its potential applications.

Moreover, our findings reveal that the aqueous alcoholic extract from *T. inodorum* herb exhibited pronounced activity against *Staphylococcus aureus* and inhibited the growth of *Escherichia coli* and *Candida albicans* in comparison to the control, a 70% ethanol solution. Furthermore, in vitro studies demonstrated that the aqueous alcoholic extract from *T. inodorum* herb exhibited significant anti-inflammatory activity, which was comparable to that of diclofenac. In addition, we demonstrated its anticoagulation, anti-aggregation, and antioxidant activities, thereby emphasizing its diverse potential for therapeutic applications.

By identifying the flavonoid composition and bioactive properties of the aqueous alcoholic extract from *T. inodorum* herb, our study contributes to the understanding of its pharmacological significance and paves the way for its potential use in pharmacy and medicine.

## Figures and Tables

**Figure 1 plants-13-01629-f001:**
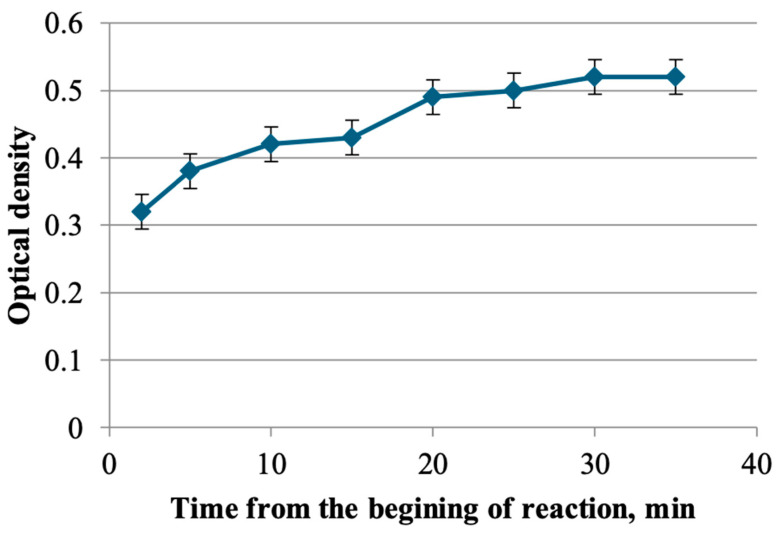
The results of the selection of conditions for the complexation reaction of flavonoids with aluminum chloride.

**Figure 2 plants-13-01629-f002:**
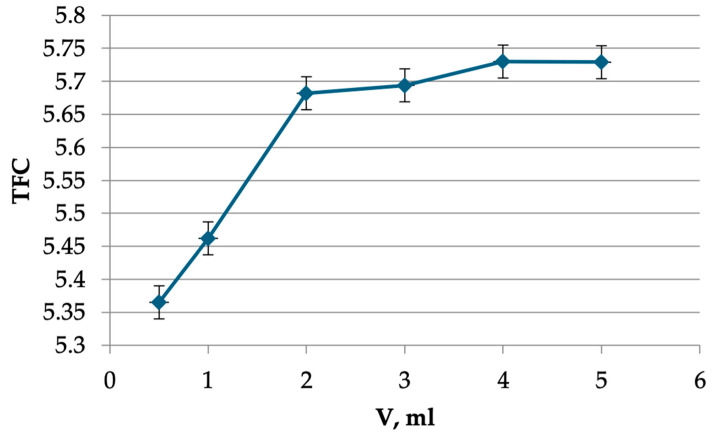
The dependence of the TFC on the volume of AlCl_3_ solution.

**Figure 3 plants-13-01629-f003:**
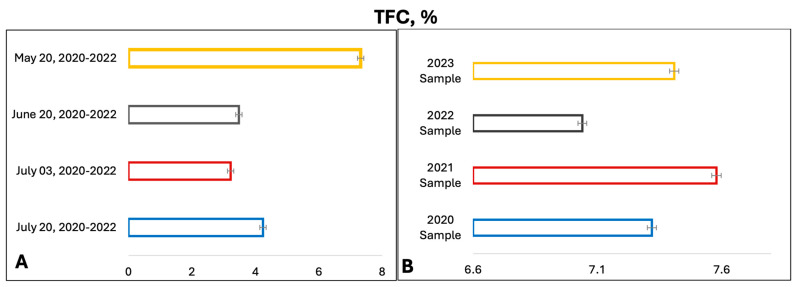
TFC in *T. inodorum* herb (**A**) depending on the growing season and (**B**) depending on the year of harvesting (note: n = 5; *p* = 0.95 (average values)).

**Figure 4 plants-13-01629-f004:**
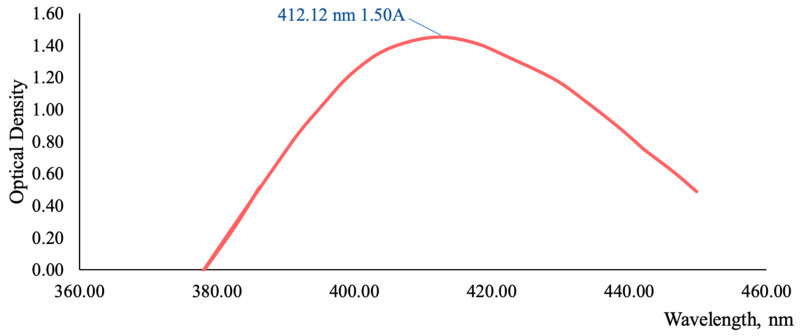
The absorption spectrum of the *T. inodorum* herb flavonoid complex (2023 sample) with aluminum chloride.

**Figure 5 plants-13-01629-f005:**
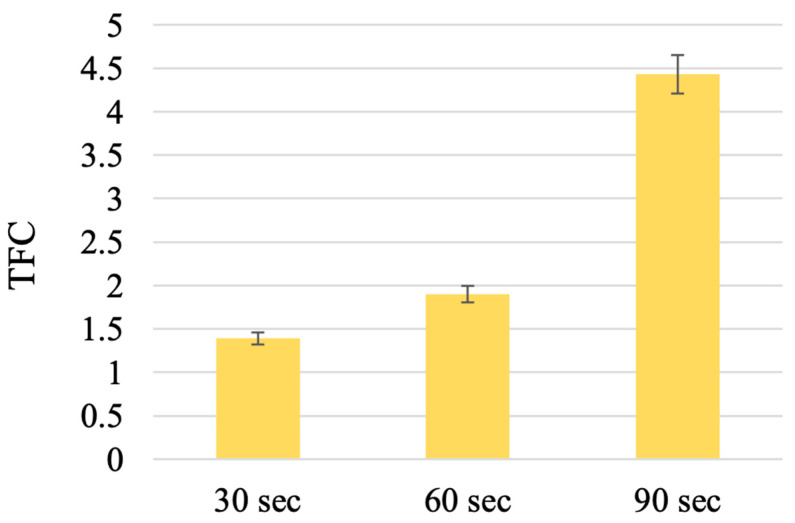
The TFC extracted by sonication with ultrasound for different times. (The results are expressed as mg REq/100 g D.W.; TFC, total flavonoid content.). Note: n = 5; *p* = 0.95.

**Figure 6 plants-13-01629-f006:**
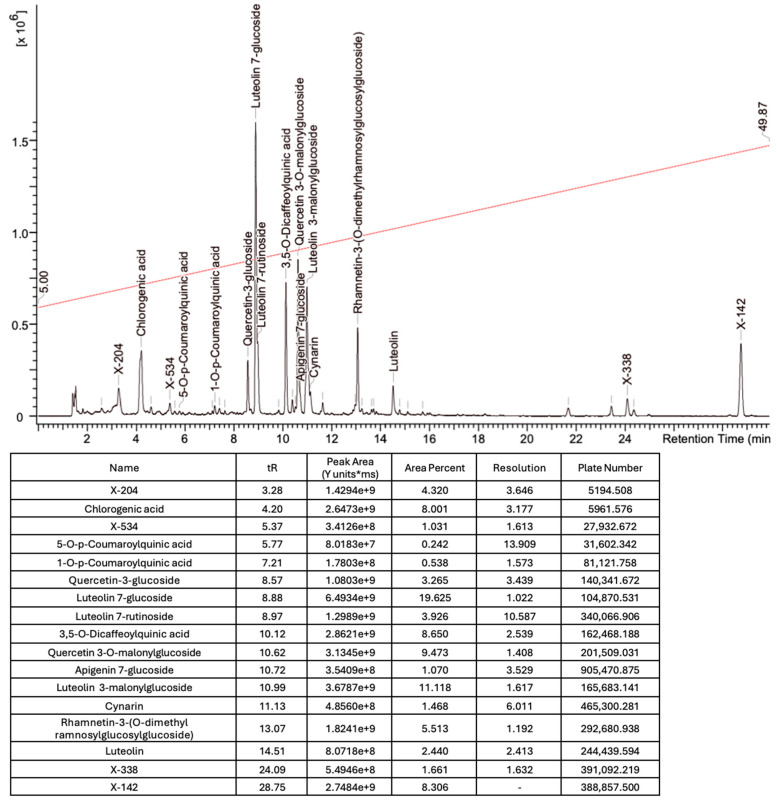
A chromatogram of the ETIH at 260 nm.

**Figure 7 plants-13-01629-f007:**
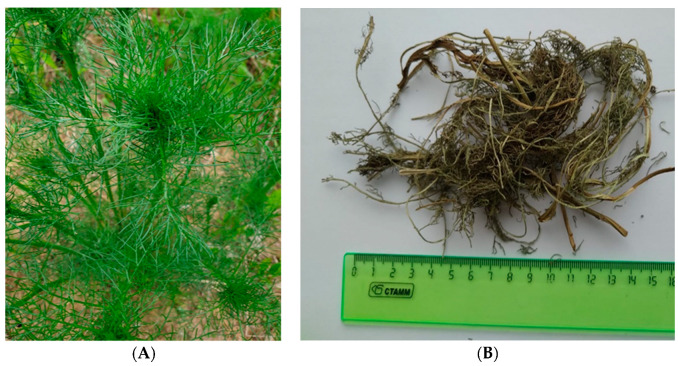
Appearance of fresh TIH harvested before flowering (**A**); dried herb (**B**).

**Table 1 plants-13-01629-t001:** The results of the selection of conditions for the extraction of flavonoids.

Concentration of Ethyl Alcohol, %	Weight of Raw Materials, g: Volume of Extractant, mL	Extraction Time, min	TFC, %
Parameter—extractant concentration			
20	1:200	20	3.844 ± 0.003
40			4.285 ± 0.001
50			4.845 ± 0.003
60			4.901 ± 0.002
70			5.025 ± 0.003
95			3.104 ± 0.003
Parameter—ratio of raw material mass, g, and extractant volume, mL			
70	1:50	40	5.424 ± 0.002
	1:100		5.564 ± 0.003
	1:150		5.684 ± 0.002
	1:200		5.681 ± 0.003
Parameter—extraction time, min			
70	1:150	20	5.025 ± 0.003
		30	5.165 ± 0.002
		40	5.682 ± 0.003
		50	5.591 ± 0.004
		60	5.426 ± 0.001

Note: n = 5; *p* = 0.95.

**Table 2 plants-13-01629-t002:** Phenolic compounds of *T. inodorum* herb.

Name	tR	Mol. Weight	UV Absorption Wavelength, nm	Structure
Chlorogenic acid	4.2	354	326	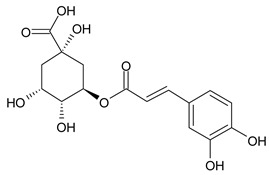
5-*O-p*-coumaroylquinic acid	5.77	338	312	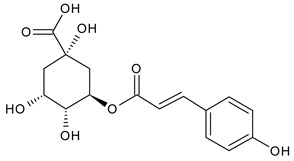
1-*O-p*-coumaroylquinic acid	7.21	338	307	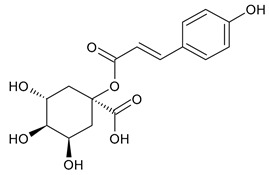
Quercetin-3-*O*-glucoside	8.57	464	370	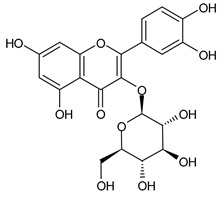
Luteolin-7-*O*-glucoside	8.88	448	348	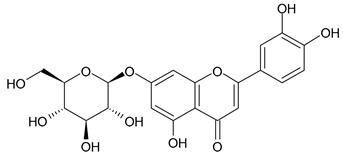
Luteolin-7-*O*-rutinoside	8.97	594	349	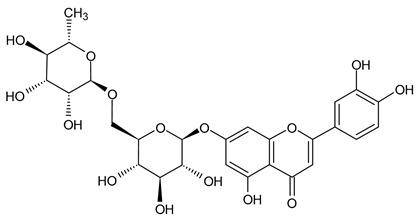
3,5-*O*-dicaffeoylquinic acid	10.12	516	327	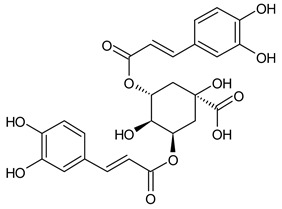
Quercetin-3-*O*-malonylglucoside	10.62	550	370	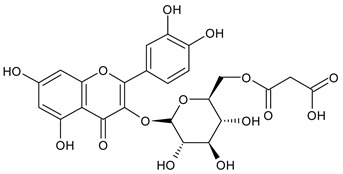
Apigenin-7-*O*-glucoside	10.72	432	344	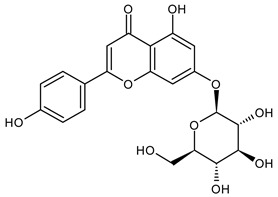
Luteolin-3-malonylglucoside	10.99	534	348	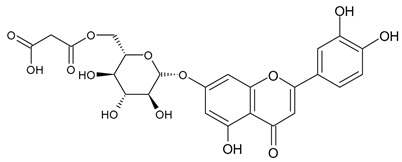
Tsinarin	11.13	516	329	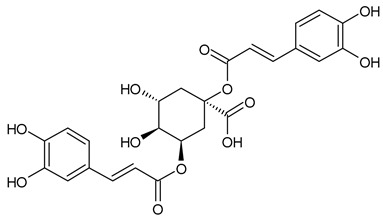
Rhamnetin-3-(*O*-dimethyl ramnosylglucosy glucoside)	13.07	814	370	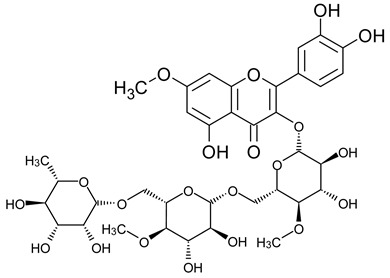
Luteolin	14.51	286	349	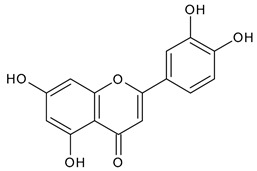

**Table 3 plants-13-01629-t003:** Antimicrobial activity of ETIH (n = 5; *p* < 0.05).

Sample	Zone of Inhibition *St. aur.*, mm	Zone of Inhibition *C. alb.*, mm	Zone of Inhibition *E. coli*, mm
ETIH	22 ± 2	12 ± 1	14 ± 2
Alcohol 70%	6 ± 1	5 ± 1	7 ± 2

Note: Data are significant in comparison with control and heparin at *p* < 0.05; n = 6.

**Table 4 plants-13-01629-t004:** Effects of sodium heparin and *T. inodorum* on parameters of plasma hemostasis.

Sample	APTT Change, % to Control	Change in PT, % to Control	Fibrinogen, % to Control
ETIH solution	+7.1 (7.3 ± 0.1)	0.0 (0)	0.0 (0)
Heparin sodium	+20.3 (20.6 ± 0.3)	0.0 (0)	0.0 (0)

Note: Data are significant in comparison with control and heparin at *p* < 0.05; n = 6.

**Table 5 plants-13-01629-t005:** Effect of *T. inodorum* and comparison drugs on platelet aggregation indicators.

Sample	Latent Period, % to Control	Maximum Amplitude, % to Control	Aggregation Rate, % to Control	Time to Reach MA, % to Control	Disaggregation, % to Control
ETIH solution	+4.2 (4.8 ± 0.2) ^††,#^	−6.5 (6.7 ± 0.1) *^,††,#^	−9.6 (10.8 ± 0.2) *^,††^	+9.6 (10.2 ± 0.3) *^,†,#^	0.0 (0) ^††^
Acetylsalicylic acid	−2.1 (1.9 ± 0.1) ^††^	−13.7 (13.6 ± 0.3) *^,††^	−10.5 (10.0 ± 0.2) *^,††^	+10.5 (10.2 ± 0.1) *^,††^	0.0 (0.0–0.0) ^††^
Pentoxifylline	+32.4 (28.7–35.6) *	−48.4 (42.7–56.5) **	−34.9 (34.1 ± 0.5) **	+ 32.1 (31.1 ± 0.4) **	13.6 (13.8 ± 0.2) **

Note: The latency period is presented for collagen-induced platelet aggregation, and the remaining parameters are for ADP-induced platelet aggregation. * *p* ≤ 0.05 and ** *p* ≤ 0.001 in comparison with the control; ^†^ *p* ≤ 0.05 and ^††^ *p* ≤ 0.001 in comparison with pentoxifylline; and ^#^ *p* ≤ 0.05 in comparison with acetylsalicylic acid; n = 6.

**Table 6 plants-13-01629-t006:** Chemiluminescence indicators on model systems for generation of reactive oxygen species (I) and lipid peroxidation (II), % of control.

Sample	Model	Light Sum	Flash
ETIH solution	I	−16.1 (16.4 ± 0.3) *^,a^	−11.4 (12.9 ± 0.4) *^,a^
	II	−32.6 (31.2 ± 0.4) **^,a^	−7.2 (9.8 ± 0.2) *^,a^
Ascorbic acid	I	−84.5 (83.6 ± 0.5) **	−91.7 (88.9 ± 0.7) **
	II	−78.1 (75.9 ± 0.5) **	−86.8 (86.5 ± 0.5) **

Note: The median and interquartile range based on the results of 6 measurements are given. * *p* ≤ 0.05 and ** *p* ≤ 0.001 in comparison with the control; ^a^ *p* ≤ 0.05 in comparison with ascorbic acid.

**Table 7 plants-13-01629-t007:** Anti-inflammatory activity of ETIH.

Sample	Paw Diameter (mm)			Severity of Inflammatory Reaction, %		Suppression of Edema at Peak of Inflammation, %
	0 h	4 h	24 h	4 h	24 h	4 h
Control	2.9 (3.3 ± 0.1)	4.7 (4.6 ± 0.1) ^a^	4.1 (3.8 ± 0.3) ^a^	62.07	41.38	-
ETIH solution	2.7 (2.8 ± 0.1)	3.3 (3.3 ± 0.1) *	3.1 (3.2 ± 0.1) *	22.22 ± 3.56	14.81 ± 2.15	66.67
Diclofenac sodium, 10 mg/kg	3.0 (2.9 ± 0.2)	3.5 (3.6 ± 0.2) *^,a^	3.2 (3.2 ± 0.1) *	16.67 ± 2.78	6.67 ± 1.05	72.22

Note: The median and interquartile range based on the results of 10 measurements are given. * *p* ≤ 0.05 in comparison with the control at the corresponding time; ^a^ *p* ≤ 0.05—0 h vs. 4 h and 24 h.

## Data Availability

Upon request, the data will be made available from the corresponding author. The data are not publicly, as they form part of a PhD dissertation. They will be made available after the dissertation has been successfully defended at the Dissertation Committee portal.

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
