# Peer review of "The Development of a Method for Obtaining Tripleurospermum inodorum (L.) Sch. Bip. Herb Extract Enriched with Flavonoids and an Evaluation of Its Biological Activity"

_plants, 2024, doi:10.3390/plants13121629_

Round 1

Reviewer 1 Report (Previous Reviewer 1)

Comments and Suggestions for Authors

ACCEPT

Author Response

Dear Reviewer:

Thanks a lot for your assistance to increase readability of our submission.

Sincerely

Dr. Alex Vetcher

Reviewer 2 Report (Previous Reviewer 2)

Comments and Suggestions for Authors

The manuscript by Marakhova et al. describes the chemical characterization of a hydroalcoholic extract from T. inodorum. However, several relevant issues make this study difficult to understand and, consequently, asses its merit and significance.

Initially, I found the manuscript inadequately organized and several passages challenging to follow. The abstract needs to be rewritten to focus on the main important findings, using a structured organization of an abstract. This problem is similar to the introduction (no precise aim and scope are presented since the combination of the flavonoid extraction optimization and the bioassays are challenging to follow) and the M&M section, whose description is highly confusing. For instance, there is no proper explanation for the qualitative and quantitative analysis of the phenolic compounds. Indeed, it seems that was not adequately performed. In addition, the results are repetitive and highly descriptive, but the main findings are not presented in depth. The method for flavonoid quantification is preliminary and has several interferences in a leaf-derived extract. The compound identification is not adequately described since low-resolution MS is not enough to provide a reliable identification, and worse if several isomers can occur (e.g., coumaroylquinic acid isomers). In addition, the activity of the extract is poor in the different bioassays, and no apparent effect of the flavonoid optimization was observed, so the aim and scope of this manuscript are called into question. The discussion summarizes results, and a correct comparison with previous studies is not provided. A similar condition is found in the conclusions.

Comments on the Quality of English Language

An editing service for this manuscript is therefore recommended since there are several important language issues.

Author Response

Reply to Reviewer 2

Dear Reviewer,

We would like to thank you for reviewing our manuscript. We are grateful for your time and review comments on our manuscript. Thank you for your constructive comments and positive feedback. We have analyzed your comments carefully and made revision to our manuscript which we hope to meet with your approval.

Here are our point-by-point answers regarding your comments:

  1. The abstract needs to be rewritten to focus on the main important findings, using a structured organization of an abstract.

The abstract was revised

  1. This problem is similar to the introduction (no precise aim and scope are presented since the combination of the flavonoid extraction optimization and the bioassays are challenging to follow)

The introduction was revised according to the comments

  1. The M&M section, whose description is highly confusing. For instance, there is no proper explanation for the qualitative and quantitative analysis of the phenolic compounds. Indeed, it seems that was not adequately performed.

The M&M section includes both qualitative and quantitative analysis – for quantitative analysis – sections 4.3-4.4; qualitative analysis – section 4.5; and biological activity – sections 4.6-4.8

  1. In addition, the results are repetitive and highly descriptive, but the main findings are not presented in depth. The method for flavonoid quantification is preliminary and has several interferences in a leaf-derived extract. The compound identification is not adequately described since low-resolution MS is not enough to provide a reliable identification, and worse if several isomers can occur (e.g., coumaroylquinic acid isomers). In addition, the activity of the extract is poor in the different bioassays, and no apparent effect of the flavonoid optimization was observed, so the aim and scope of this manuscript are called into question. The discussion summarizes results, and a correct comparison with previous studies is not provided.

In our study, we employed standard samples to enhance the accuracy and reliability of our results. By comparing the retention times of the peaks in our analysis with those of the standard samples, we were able to confirm the identity of the compounds. In quantification process, we also utilized standard samples to ensure the precision and validity of our experimental measurements.

In the discussion section, we encountered challenges due to the limited availability of studies specifically focusing on TIH. However, we tried drawing comparisons with existing research on flowers or other plants belonging to the genus Tripleurospermum.

  1. A similar condition is found in the conclusions.

The conclusion was revised

Please let us know if we can do something else to increase the readability of our submission.

Regards

Dr. Alexandre A Vetcher

Reviewer 3 Report (Previous Reviewer 3)

Comments and Suggestions for Authors

Regrettably, the paper still lacks a significant amount of the requested information (referenced in the attachment) compared to the initial draft. A majority of the posed questions remain unanswered or the responses provided are inadequate. Specifically, questions 1, 2, 3, 7, 19, and 20 have not been addressed. It’s unclear which part of the plant is utilized in each subsequent bioassay. Furthermore, the quantities of samples and standards used in the bioassays are presented in a confusing manner, making it difficult to comprehend the significance of the results. I must express my apologies, but in its current state, both in terms of content and presentation, the work does not meet the standards for publication, in my view

Comments on the Quality of English Language

The quality of scientific language is low

Author Response

Reply to Reviewer 3

Dear Reviewer,

Thank you for reviewing our manuscript. We are grateful for your time and constructive comments. We have analyzed them, revised our manuscript accordingly, and hope that our revised manuscript meets with your approval.

Here are our point-by-point answers regarding your comments:

11) A study regarding the pharmacological activity of an extract should contemplate at least the experimental designs aimed to ascertain a dose response activity of the studied extracts. In the case of this work, I warmly suggest changing the title inBioactivity of….”

Corrected accordingly

22) Please, add a dot point after Bip (Bip.)

Corrected accordingly

Introduction

33) Please, modify the text in accordance to the comment 1)

Corrected accordingly, but, please, pay attention that there is no comment 1) in the review.

77)      Please, clearly state all the units of measurements, for example in line 83: the amount of flavonoids in terms of rutin (R) equivalent (Req) and absolutely dry weight (D.W.) of raw materials (mg Req/100gr D.W.)

Corrected accordingly. Most of measurements are in %

Discussion.

19) The results and the cited bibliography should be more extensively compared and discussed. The discussion should not contain technical information regarding the experiments (see lines 395-398 as an example).

In the discussion section, we encountered challenges due to the limited availability of studies specifically focusing on TIH. However, we tried drawing comparisons with existing research on flowers or other plants belonging to the genus Tripleurospermum.

 20) The “Conclusions” paragraph shouldn’t be a repetition of the Abstract, or a brief summary of the results. It should most of all explains the implications of this study in the field and propose future applications.

Corrected accordingly

Please let us know if we can do something else to increase the readability of our submission.

Regards

Dr. Alexandre A Vetcher

Reviewer 4 Report (Previous Reviewer 5)

Comments and Suggestions for Authors

Table 1

As I mentioned before, this table in unnecessary. Please delete it and describe in text, such as 0min: %B 5, 30min: %B 50, etc.

Figure 4A and 4B

As I mentioned before, these two figures should be presented in same style with same axis.

Table 4

The chemical structure of compounds should be indicated with same size.

Author Response

Reply to Reviewer 4

Dear Reviewer,

Thank you for reviewing our manuscript. We are grateful for your time and constructive comments. We have analyzed them, revised our manuscript accordingly, and hope that our revised manuscript meets with your approval.

Here are our point-by-point answers regarding your comments:

  1. Table 1. As I mentioned before, this table in unnecessary. Please delete it and describe in text, such as 0min: %B 5, 30min: %B 50, etc.

Corrected accordingly

  1. Figure 4A and 4B. As I mentioned before, these two figures should be presented in same style with same axis.

Corrected accordingly

  1. Table 4. The chemical structure of compounds should be indicated with same size.

Corrected accordingly

Please let us know if we can do something else to increase the readability of our submission.

Regards

Dr. Alexandre A Vetcher

Round 2

Reviewer 2 Report (Previous Reviewer 2)

Comments and Suggestions for Authors

The authors addressed my comments and suggestions adequately, so the manuscript improved in quality and content. Consequently, it can be accepted in its current form.

Author Response

Dear Reviewer,

thank you for your assistance in the incrementing quality of our submission

Regards

Dr. Alex Vetcher

Reviewer 3 Report (Previous Reviewer 3)

Comments and Suggestions for Authors

Despite significant improvements, in my opinion the manuscript is presented in a manner that is still far from being accepted for publication. This is a list of issues that haven't been addressed.

1)     A study regarding the pharmacological activity of an extract should contemplate at least the experimental designs aimed to ascertain a dose response activity of the studied extracts. In the case of this work, I warmly suggest changing the title in “Bioactivity of….”

Corrected accordingly

Please correct also lines 299-330 and 517.

Despite good conduct, this study lacks the strength to determine any pharmacological activity. The only activity that the authors can suggest is a biological one. The distinction is that authors must show that their extract has a dose-response activity, or/and a statistically significant correlation between their phytochemical and pharmacological parameters.

2) Please, add a dot point after Bip (Bip.)

Corrected accordingly

Please, correct it also in the title

Introduction

33) Please, modify the text in accordance to the comment 1)

Corrected accordingly, but, please, pay attention that there is no comment 1) in the review.

I apologize for the error in my numbering.

77)      Please, clearly state all the units of measurements, for example in line 83: the amount of flavonoids in terms of rutin (R) equivalent (Req) and absolutely dry weight (D.W.) of raw materials (mg Req/100gr D.W.)

Corrected accordingly. Most of measurements are in %

I warmly recommend you to express the mean values in a conventional manner. For instance, you have to include the sentence "Results are expressed as rutine equivalent (R Eq) / 100 g of fresh weight (F.W.) (mg REq/ 100 g F.W.)" at the conclusion of paragraph 4.3. Technique development for analyzing total flavonoid content (or 100 g of dry weight D.W, in case you weighted the dried material, this is also not clear. If the samples were dried, how was the drying process carried out?)

In table 1 (and also in figure 5), please change "Total flavonoids in terms of rutine, %" to "TFC" and add to the footnote (to the caption, for fig. 5) the sentence "Results expressed as mg REq/ 100 g F.W.; different letters in the same column represent statistically different results at p < 0.XX; TFC, total flavonoids content".

In the text, for example line 37, it will be 7.65 ± 0.03 mg REq/ 100 g F.W.

Kindly review your manuscript and make the necessary changes based on this suggestion. Have a look on paper like this https://doi.org/10.1038/s41598-020-77991-2  or this doi: 10.3390/ijms130810257 as examples

 Discussion.

19) The results and the cited bibliography should be more extensively compared and discussed. The discussion should not contain technical information regarding the experiments (see lines 395-398 as an example).

In the discussion section, we encountered challenges due to the limited availability of studies specifically focusing on TIH. However, we tried drawing comparisons with existing research on flowers or other plants belonging to the genus Tripleurospermum.

OK

 20) The “Conclusions” paragraph shouldn’t be a repetition of the Abstract, or a brief summary of the results. It should most of all explains the implications of this study in the field and propose future applications.

Corrected accordingly

OK

- The statistical analyses for the means presented in Tables 1, 3, and 4 and Figures 1, 2, 3, and 5 should be carried out and appropriately showed in the table/figure; each mean should be presented with its error (or deviation) standard rather than with its range values. (Tables 4 through 7) (Table 4 in pag 7 should be Table 2)

- Put the word "extraction" in the titles of paragraphs 4.3 and 4.4 since they both also deal with the process of preparation of the extracts rather than just its analysis.

- Please specify which standard solutions (and their concentrations) are mentioned in paragraph 4.5, lines 397.

- Procedure descriptions go in the Materials and Methods section and should not be found in the Results section. Lines 162-174 and 197-198 should be moved to Materials and Methods.

- Please include the quantification methodologies for the tannins and polysaccharide contents discussed in lines 189–191 as there is no mention of them in the materials and methods section.

- When referring to an optical density, please always include the acquisition nm (see formula 1)

- Adding a list of abbreviations is warmly suggested

- You can use the acronyms in all parts of the text, excluding the abstract, captions, and footnotes.

Comments on the Quality of English Language

Poor attention to scientific language,  unconventional way to present results

Author Response

2024-05-29

Reply on review

Dear Reviewer:

Thank you so much for your careful imput in the increament of the quality of our submission. As about your comments – let me respond on them in the order in your review:

Despite significant improvements, in my opinion the manuscript is presented in a manner that is still far from being accepted for publication. This is a list of issues that haven't been addressed.

1)     A study regarding the pharmacological activity of an extract should contemplate at least the experimental designs aimed to ascertain a dose response activity of the studied extracts. In the case of this work, I warmly suggest changing the title in “Bioactivity of….”

Corrected accordingly

Please correct also lines 299-330 and 517.

Despite good conduct, this study lacks the strength to determine any pharmacological activity. The only activity that the authors can suggest is a biological one. The distinction is that authors must show that their extract has a dose-response activity, or/and a statistically significant correlation between their phytochemical and pharmacological parameters.

Corrected accordingly

2) Please, add a dot point after Bip (Bip.)

Corrected accordingly

Please, correct it also in the title

Corrected accordingly

Introduction

33) Please, modify the text in accordance to the comment 1)

Corrected accordingly, but, please, pay attention that there is no comment 1) in the review.

I apologize for the error in my numbering.

77)      Please, clearly state all the units of measurements, for example in line 83: the amount of flavonoids in terms of rutin (R) equivalent (Req) and absolutely dry weight (D.W.) of raw materials (mg Req/100gr D.W.)

Corrected accordingly. Most of measurements are now in %

I warmly recommend you to express the mean values in a conventional manner. For instance, you have to include the sentence "Results are expressed as rutine equivalent (R Eq) / 100 g of fresh weight (F.W.) (mg REq/ 100 g F.W.)" at the conclusion of paragraph 4.3. Technique development for analyzing total flavonoid content (or 100 g of dry weight D.W, in case you weighted the dried material, this is also not clear. If the samples were dried, how was the drying process carried out?)

Corrected accordingly

In table 1 (and also in figure 5), please change "Total flavonoids in terms of rutine, %" to "TFC" and add to the footnote (to the caption, for fig. 5) the sentence "Results expressed as mg REq/ 100 g F.W.; different letters in the same column represent statistically different results at p < 0.XX; TFC, total flavonoids content".

Corrected accordingly

In the text, for example line 37, it will be 7.65 ± 0.03 mg REq/ 100 g F.W.

Corrected accordingly

Kindly review your manuscript and make the necessary changes based on this suggestion. Have a look on paper like this https://doi.org/10.1038/s41598-020-77991-2  or this doi: 10.3390/ijms130810257 as examples

Corrected accordingly

 Discussion.

19) The results and the cited bibliography should be more extensively compared and discussed. The discussion should not contain technical information regarding the experiments (see lines 395-398 as an example).

Corrected accordingly

In the discussion section, we encountered challenges due to the limited availability of studies specifically focusing on TIH. However, we tried drawing comparisons with existing research on flowers or other plants belonging to the genus Tripleurospermum.

OK

 20) The “Conclusions” paragraph shouldn’t be a repetition of the Abstract, or a brief summary of the results. It should most of all explains the implications of this study in the field and propose future applications.

But rice. 1 and 2, in our opinion, do not need correction, since they do not reflect exact values, but only demonstrate conditions when the optical density or flavonoid content does not change.

OK

- The statistical analyses for the means presented in Tables 1, 3, and 4 and Figures 1, 2, 3, and 5 should be carried out and appropriately showed in the table/figure; each mean should be presented with its error (or deviation) standard rather than with its range values. (Tables 4 through 7) (Table 4 in pag 7 should be Table 2)

Corrected accordingly. But fig. 1 and 2, in our opinion, do not need correction, since they do not reflect exact values, but only demonstrate conditions when the optical density or flavonoid content does not change.

- Put the word "extraction" in the titles of paragraphs 4.3 and 4.4 since they both also deal with the process of preparation of the extracts rather than just its analysis.

Corrected accordingly

- Please specify which standard solutions (and their concentrations) are mentioned in paragraph 4.5, lines 397.

Corrected accordingly

- Procedure descriptions go in the Materials and Methods section and should not be found in the Results section. Lines 162-174 and 197-198 should be moved to Materials and Methods.

Corrected accordingly

- Please include the quantification methodologies for the tannins and polysaccharide contents discussed in lines 189–191 as there is no mention of them in the materials and methods section.

Corrected accordingly. It was a technical mistake

- When referring to an optical density, please always include the acquisition nm (see formula 1)

Corrected accordingly

- Adding a list of abbreviations is warmly suggested

Corrected accordingly

- You can use the acronyms in all parts of the text, excluding the abstract, captions, and footnotes.

Corrected accordingly

Please let us know, if we can do something else to improve the quality of our submission.

Sincerely

Dr. Alex Vetcher

This manuscript is a resubmission of an earlier submission. The following is a list of the peer review reports and author responses from that submission.

Round 1

Reviewer 1 Report

Comments and Suggestions for Authors

The overall clarity of the images in the manuscript is not enough

Figure 2 ETIHs, obtained from raw materials shared in different years, readers are unable to understand what the image is intended to convey.

The overall quality of the manuscript is poor and requires careful editing and improvement before it can be published.

Comments on the Quality of English Language

The overall quality of the manuscript is poor and requires careful editing and improvement before it can be published.

Reviewer 2 Report

Comments and Suggestions for Authors

The reviewed manuscript explores the chemical and biological activity evaluation of Tripleurospermum inodorum. Although the content can be considered interesting and informative, several concerns arise regarding the structure, scope, paradigm, organization, and findings, which limit further consideration.

Major Concerns:

  1. The study appears preliminary rather than comprehensive, relying on hydroalcoholic preparation involving preliminary chemical characterization and unjustified biological activity evaluations. A more robust justification is needed.
  2. The aim and scope lack clarity, and there is insufficient justification for employing common and preliminary methods to study the plant material.
  3. Compound identification based solely on m/z values of adducts is inadequate. A more rigorous approach is essential for accurate identification.
  4. The Materials and Methods section requires improvement, lacking informative details crucial for reproducibility despite containing numerous details.
  5. The reported flavonoid content (approximately 7%) is unusually high and warrants clarification or validation.
  6. Biological activity-related results are not adequately explained, and the test extract shows poor activity in performed assays. Further clarification and contextualization are necessary.
  7. The discussion section is overly descriptive and speculative, not properly comparing with previous studies. The organizational structure needs refinement for better coherence.
  8. The conclusion section, while summarizing results, would benefit from a rewrite to present conceptual findings from a mechanistic standpoint, providing a more cohesive ending.

Other Points:

  1. The abstract requires restructuring. Currently, it appears to merely declare, in a brief and unclear manner, the data presented in the article. It would be more effective if the abstract succinctly and clearly generalizes the article's content. The goal is to create a shorter and more concise abstract that effectively communicates the article's essence.
  2. Line 24: Why two plants are mentioned here and introduction (line 34). Are they related to synonyms? Challenging species to be classified? This information must be clarified to the readers.
  3. The introduction must be improved since it is laconically developed.

4.    Lines 53-57: The aim and scope of the study is not clear. Revise these lines for improved clarity regarding the study's aim and scope.

  1. Line 55: ETIH must be defined in the introduction. It is defined in the abstract, but it must also defined in the first use in the manuscript body.
  2. Line 60: Is ETIH related to the plant or the plant extract? Revise.
  3. Line 60: Why harvested before flowering? Any reason for that? This information must be added to the manuscript.
  4. Line 62: Why 1 g? Was it enough to perform the entire study? In addition, what does “exactly” mean using only one significant figure? There are no significant figures to define the exact amount.
  5. Figure 2: This picture is not related to any information in the text regarding different extracts obtained in different years.
Comments on the Quality of English Language

The manuscript requires a comprehensive and meticulous review that focuses on organizational, grammatical, stylistic, and potential typographical issues throughout the document. An editing service is recommended.

Reviewer 3 Report

Comments and Suggestions for Authors

The Article “Pharmacological activity of the Tripleurospermum inodorum (L.) Sch. Bip herb extract from Anna Marakhova, Alexander Elapov, Vera Zhylkina, Nadezhda Sachivkina, Alexander Samorodov, Kira Pupykina, Irina Krylova and Alexandre A. Vetcher proposes the first comprehensive report on the qualitative and quantitative composition of the serum bacterial activity of Tripleurospermum inodorum harvested before flowering. The authors find that the flavonoid content can reach up to 7% and showed the presence of 13 phenolic compounds The hydro-alcoholic extract from T. inodorum herb showed activity against Staphylococcus aureus, and inhibited the growth of Escherichia coli and Candida albicans. It also exhibited an anti-inflammatory activity comparable to diclofenac, and anticoagulation, antiaggregation, and antioxidant activities.

Although the authors in this work have made a great effort to analyze the extracts and characterize their biological activity, also with the use of animal and human samples, however the manuscript cannot be accepted in its present form, especially because the Material and Method and Results sections are unfortunately presented poorly and often in an unconventional manner. The discussion also is very poor and the conclusions need to be rewritten.

Here is a partial list of some of the issues I encountered, starting from the title:

 Title

11)      A study regarding the pharmacological activity of an extract should contemplate at least the experimental designs aimed to ascertain a dose response activity of the studied extracts. In the case of this work, I warmly suggest to change the title in “Bioactivity of….”

22)      Please, add a dot point after Bip (Bip.)

Introduction

33)      Please, modify the text in accordance to the comment 1)

Abstract

44)      Lines 28-29 seem a repetition of lines 25-26

Material and Methods

55)      Paragraph 2.1 is about the sample extraction procedure and not about the plant collection. Please, rename this paragraph and add a proper “Plant collection” paragraph as it is almost mandatory to specify where the plants were collected and their growing conditions.

66)      An aqueous-alcoholic extract of flowers of T. inodorum is reported in table 3: please add the extraction procedure of this plant material, or add the bibliographic reference(s) if the values come from the literature.

77)      Please, clearly state all the units of measurements, for example in line 83: the amount of flavonoids in terms of rutin (R) equivalent (Req) and absolutely dry weight (D.W.) of raw materials (mg Req/100gr D.W.)

8) Line 91: what is the sample? Is it 1 gr (fresh weight) of Tripleurospermum inodorum? Thus, paragraph 2.2.1. UPLC/MS -analysis should be “Extraction and UPLC/MS analysis” Please, clarify

9) 2.2.2. Determination of the content of total flavonoids: sorry, but the sentence “2.0 ml of solution A of the test solution” doesn’t make any sense to the reader and make your experiments impossible to replicate. Maybe you only need to state the commercial kit that you have utilized, or/and the bibliographic reference(s) of your protocol. In this case, you can even summarize the paragraph. Please, have a look through the Material and Methods section and make the changes according to this suggestion.

10) Paragraphs 2.3 and 2.5: please clearly state the dilution and the uL or ml administered both of the extract and the comparison drugs; briefly report this information in the caption of each figure or graphic or in the note of the tables

Results

11) Please, correctly report every time the units of measurement thought the text

Every table, graphics or figure should “stand alone”, in particular:

12) Graphics: please, add in all abscissa axes the units of measurement (i.e. in fig. 4 should be mg Req/100gr D.W.?); add in all columns (either vertical or horizontal) the standard error bars; add in the captions the number of replicates (n=?) and the level of significance (p= 0.05?).

13) Please, briefly report the information on the dilution of the extract and the ul or ml administered in a note from table 3 to table 7. For example in table 3: 25? ul ETIH (dilution 1:?). Also always report the number of replicates (n=?); in table 7 you could eliminate the column “number of animals.”

 14) Table 3, 6, 7: replace “object of study”, “Test substance” and “A drug” with “sample”

15) Table 4 and 5: what does “Me (0.25-0.75)” stand for?

 16) Figure 3 and lines 328-336 in my opinion are redundant and should be eliminated. Rutin is a common standard for the quantitation of flavonoids.

 17) The data on the content of total flavonoids, depending on the growing season, cannot be accepted if the year of their collection is only one, because the data of a single year of collection strongly depends on the season variability of that specific year. You need not less than three years of collection, for plant samples collected from open fields.

18) It is not clear to me the reason why anticoagulation, antiaggregation, antioxidant, and anti-inflammatory activities of are presented only for ETIH. The worst performance in antimicrobial activity of the aqueous-alcoholic extract of flowers of T. inodorum compared to ETIH alone does not justify having it been discarded

19) Discussion

The results and the cited bibliography should be more extensively compared and discussed. The discussion should not contain technical information regarding the experiments (see lines 395-398 as an example).

 20) The “Conclusions” paragraph shouldn’t be a repetition of the Abstract, or a brief summary of the results. It should most of all explains the implications of this study in the field and propose future applications.

21) Please, format all references according to the journal style; n. 32 is not complete

Comments on the Quality of English Language

The text is sometimes not clear, somewhat unconventional and not enough comprehensible

Reviewer 4 Report

Comments and Suggestions for Authors

Up to the extent of available knowledge, in a typical SBA assay (Serum Bactericidal Assay) , a serial dilutions of sera are incubated with target bacterial strains. The SBA reaction mixture will be plated on agar and surviving bacterial colony-forming units (CFU) will be counted at each serum dilution and luminescence-based SBA (L-SBA) methods detect surviving bacteria by measuring their ATP. Authors mention SBA assay (bacterial, not bactericidal) (line 15) saying "The current study is the first comprehensive report on the qualitative and quantitative composition of the SBA (serum bacterial activity) of Tripleurospermum inodorum", but in fact they determine the antimicrobial activity using classical disk diffusion tests. This should be amended.

They use blood from donors, but not for antimicrobial tests.

For the preparation of TIH extracts they do not detail how they harvest and dry the plants. Methods go directly to extraction methodology.

Some methodology paragraphs are incorrect like "The resulting solution was evaporated on a rotary solvent to remove ethyl alcohol ....."(line 76).

Line 229 oral administration should be detailed. (administered forcibly.....)

Since they claim that "pharmacological activity was established for the first time for ETIH harvested before flowering" details of raw material obtention (wet and dry weigth....) are specially relevant. Conclusions can be imrpoved, in some paragraphs the repeat evident infromation given before.

Comments on the Quality of English Language

Line 427 "compari-son" , there are several spelling errors like this.

Reviewer 5 Report

Comments and Suggestions for Authors

This manuscript deals with chemical composition and physiological activity of Tripleurospermum indorum extract. The authors qualitatively and quantitatively analyzed phenolic compounds on the basis of LC/MS, and evaluated antimicrobial, anti-inflammatory activities. The experiments were carried out carefully, and the results were described in detail. I think these data are useful in this field, so this study should be published with some revisions as bellow,

Material and Methods

The description in this chapter is too detailed and complicated. Please simplify it a little more

Table 1

This table in unnecessary. Please delete it and describe in text, such as 0min: %B 5, 30min: %B 50, etc.

Figure 4 and 5

Please present these two figures in same style.

Figure 5

Were there any significant differences in flavonoid content between years? Please mention.

Table 2

Please simplify the table by removing the structures.
